## [Decision Letter · Decision Letter 0]

11 Jul 2025

Dear Dr. Zhao,

Thank you for submitting your manuscript to PLOS ONE. After careful consideration, we feel that it has merit but does not fully meet PLOS ONE’s publication criteria as it currently stands. Therefore, we invite you to submit a revised version of the manuscript that addresses the points raised during the review process. 

Clarify and briefly define core concepts and abbreviations.Reframe statistical results into more narrative-friendly language.Expand slightly on theoretical contributions and implications.Prioritize clarity and flow over data density in the abstract format.

This is a highly promising and original contribution. With minor revisions aimed at improving clarity and accessibility, it will be much better positioned to engage readers across disciplinary boundaries.

We look forward to receiving your revised manuscript.

Kind regards,

Yasemin Özkent

Academic Editor

PLOS ONE

Journal Requirements:

I your submission does not contain these data, please either upload them as Supporting Information files or deposit them to a stable, public repository and provide us with the relevant URLs, DOIs, or accession numbers. For a list of recommended repositories, please see https://journals.plos.org/plosone/s/recommended-repositories.

5. Please ensure that you refer to Figure 1 in your text as, if accepted, production will need this reference to link the reader to the figure.

Reviewers' comments:

Reviewer's Responses to Questions

**Comments to the Author**

1. Is the manuscript technically sound, and do the data support the conclusions?

Reviewer #1: Yes

2. Has the statistical analysis been performed appropriately and rigorously?

Reviewer #1: Yes

3. Have the authors made all data underlying the findings in their manuscript fully available?

Reviewer #1: Yes

4. Is the manuscript presented in an intelligible fashion and written in standard English?

Reviewer #1: No

Reviewer #1: There seems to be a lack of flow in the manuscript-everything ends abruptly. I suggest revisions in the writing style, in order to make it more confortable for the reader.

Alongside, there are grammatical errors (e.g. in the 'References' section-it is mentioned as 'Refernces'!), and the whole of the manuscript should be checked again for any typographical and grammatical errors.

Each point starts with an uppercase letter after a colon, which is inconsistent with common academic style. Use lowercase letters or rephrase for fluidity

**Do you want your identity to be public for this peer review?** For information about this choice, including consent withdrawal, please see our Privacy Policy

Reviewer #1: No

---

## [Author Response · Author response to Decision Letter 1]

12 Aug 2025

I am pleased to resubmit our revised manuscript entitled “Cultural Anchoring in Cinematic Digitalization: Stratified PCA Reveals Chromatic Adaptation Regimes in Wuxia Cinema Evolution” (Manuscript number: PONE-D-25-16196) for your consideration. We have rigorously addressed all reviewer comments through significant enhancements:

Abstract Accessibility Enhancement

The abstract has been fundamentally restructured to prioritize narrative clarity over technical density. Statistical results have been transformed into descriptive interpretations (e.g., β=0.63 expressed as "significant acceleration in editing pace"), while core methodological concepts receive operational definitions at first occurrence. We implemented rigorous parameter filtering to retain only statistically significant coefficients (p<0.05), eliminating redundant metrics without compromising scientific validity.

Theoretical Framework Expansion

Three original theoretical constructs have been integrated throughout the manuscript: (1) "Chromatic sovereignty" (defined as strategic reclamation of color semantics) anchors the introduction; (2) "Parametric resistance" mechanisms via PC3 compensation are demonstrated in the discussion; (3) The paradigm shift from "technological subjugation to materiality-based sovereignty" forms the conceptual culmination in the conclusion. Each construct is empirically grounded in our analytical findings.

Terminology Standardization

All technical terminology now carries explicit definitions at point of first use: HSV components are operationally defined in the Methods section (Hue as spectral wavelength identifier, Saturation as color purity metric, Value as luminance indicator). Editing rhythm parameters are clarified with ASL representing global pacing and MSL denoting action-sequence intensity. Novel theoretical constructs include precise conceptual boundaries, particularly "parametric resilience" which denotes threshold-activated modulation systems.

Language Optimization

Substantial enhancements to linguistic fluency include: addition of 12 transitional sentences to improve logical flow, correction of grammatical inconsistencies (subject-verb agreement and prepositional usage), and strategic deconstruction of complex sentences reducing average length by 38%. These revisions were validated through professional editing by Elsevier Language Services, with certification documentation provided.

Statistical Reporting Precision

Key statistical presentations have been optimized for clarity: PC1 temporal discontinuity is now reported as ΔM=1.62 (Cohen's d=2.11) with annualized β=0.63; PCA variance allocations explicitly note PC3's 16.265% unique contribution; effect size interpretations reference established disciplinary benchmarks (η²=0.448 for PC1 exceeding cultural studies' large-effect threshold).

Data Transparency Compliance

The complete original dataset, including complete HSV parameter data and editing cadence metrics (ASL, MSL), along with detailed access instructions for the source wuxia films, is publicly archived in a GitHub repository (MIT License, file name: Accessible-video-links-and-original-datasets-of-30-wuxia-films). To ensure full analysis reproducibility, the processing code for HSV extraction (film-HSV) and editing metric calculation (film-ASL-MSL) is publicly available in separate GitHub repositories under the MIT license. Reference Standardization

The bibliography has been comprehensively updated with DOI integration where available, strict PLOS ONE formatting compliance, and removal of any retracted literature. All citations now feature complete publication metadata including publisher locations and digital object identifiers.

---

## [Editor Report · Decision Letter 1]

22 Sep 2025

Cultural Anchoring in Cinematic Digitalization: Stratified PCA Reveals Chromatic Adaptation Regimes in Wuxia Cinema Evolution

PONE-D-25-16196R1

Dear Dr. Zhao,

We’re pleased to inform you that your manuscript has been judged scientifically suitable for publication and will be formally accepted for publication once it meets all outstanding technical requirements.

Kind regards,

Yasemin Özkent

Academic Editor

PLOS ONE
---

## [Editor Report · Acceptance letter]

PONE-D-25-16196R1

PLOS ONE

Dear Dr. Zhao,

I'm pleased to inform you that your manuscript has been deemed suitable for publication in PLOS ONE. Congratulations! Your manuscript is now being handed over to our production team.

Kind regards,

on behalf of

Dr. Yasemin Özkent

Academic Editor

PLOS ONE